# Research on the development trend of global running protection equipment technology from the perspective of patent measurement

**Qiong Shen, Yao Liu** *, **Lei Qiang**

College of Basic Education, Wuxi Taihu University, Wuxi, China

* 15061116221@163.com

## Abstract

This study provides a systematic examination of global development trajectories in running protective equipment by applying large-scale patent mining and quantitative analysis to international and domestic patent data. The results identify three distinct phases in the evolution of this field: the nascent stage (1898–1970), the accumulation period (1971–1999), and the rapid expansion phase after 2000. Patent activity has been dominated by enterprises in the United States, which maintain a leading position in both volume and quality, while China has demonstrated rapid growth in filings since the early 2000s, gradually enhancing its international visibility despite a comparatively late entry. Analysis of patent validity indicates that only 34.47% of patents remain active, with more than half lapsing within a relatively short time frame. This pattern reflects limited incentives for long-term maintenance and reveals structural weaknesses in sustaining technological outputs. Furthermore, the technological evolution analysis shows a pronounced concentration of innovation in three domains—running shoes, sportswear, and knee pads—while diversification into emerging areas such as advanced materials and intelligent features remains underdeveloped. This imbalance, defined by low durability and narrow domain coverage, constrains commercialization potential and reduces the economic viability of many patented inventions. Core patent screening using a TOPSIS-based model demonstrates that high-quality patents are concentrated in a small number of enterprises and IPC subclasses, underscoring the uneven distribution of technological impact and market potential across the industry. Key technological trajectories are characterized by advances in sole stabilization and cushioning systems, optimization of textile structures and stitching techniques, integration of protective elements into apparel, and improvements in the structural and cushioning design of knee pads, with gradual progression toward material-enhanced and intelligent applications. These findings provide an evidence-based framework for understanding the global dynamics of running protective equipment technology and offer critical insights for guiding industrial upgrading, patent management strategies, and policy design aimed at fostering sustainable innovation and commercialization in this sector.

**Data availability statement:** The data underlying the findings of this study are available from the Patsnap Global Patent database (https://www.patsnap.com). Due to licensing restrictions and user privacy concerns, we cannot provide direct account access. However, the dataset is fully accessible to any researcher with Patsnap access. We have provided the detailed search strategy and retrieval steps for the dataset in the manuscript to allow replication of the data collection process.

**Funding:** This study was supported by the project Research on the Cross-border Integrated Development of Outdoor Sports Industry in the Yangtze River Delta (Grant No. 19BTY060). The sponsor is the academic advisor of the corresponding author, Prof. Liu Yao. He provided funding support only and made no intellectual or practical contribution to the study. With his explicit consent, he was not included in the author list.

**Competing interests:** The authors have declared that no competing interests exist.

## Introduction

Since the issuance of the "Several Opinions on Accelerating the Development of the Sports Industry and Promoting Sports Consumption", China's sports industry has entered a period of rapid growth, beginning in 2014 [1]. This sector has emerged as a significant driver of national economic development. However, with continuous improvements in living standards, consumer preferences have become increasingly diverse and personalized. This shift has exposed a clear supply-demand imbalance in China's sporting goods market, with the current supply of equipment no longer able to meet changing consumer expectations. The report from the 20th National Congress of the Communist Party of China emphasizes that "high-quality development is the foremost task in the comprehensive construction of a modern socialist country" [2]. This directive highlights key deficiencies within China's sports industry, particularly the inadequate capacity for scientific research and technological innovation. These limitations have resulted in many domestic products being relegated to the lower tiers of the global value chain [3]. In this context, the failure to transition from traditional manufacturing models to an innovation-driven development paradigm will severely hamper the industry's ability to align with contemporary societal demands and global trends [4]. Promoting high-quality development within the sports sector is therefore essential to addressing the supply-demand gap, enhancing industrial competitiveness, and elevating China's position within the global sports equipment market.

Running is widely regarded as a fundamental activity for promoting national fitness. However, during running, sustained mechanical stress on the limbs, combined with repetitive friction among skeletal muscles and irregular movement patterns, frequently leads to various physical injuries and conditions [5]. These include knee joint degeneration, tibial stress syndrome, plantar fasciitis, muscle spasms, and other related ailments [6]. Empirical evidence indicates that protective equipment represents the most effective strategy for both preventing and managing such issues. This approach is associated with high levels of user satisfaction, strong compliance rates, and significant protective benefits. In 2021, the revenue of running-related products in China has reached 183.12 billion yuan, showing the huge market of the industry. Nevertheless, as the manufacturing of running protective equipment is closely tied to human movement science, it must adhere to rigorous quality and technical standards, necessitating substantial research and development (R&D) efforts to drive innovation. Simultaneously, key technologies in this domain remain concentrated in a small number of countries and are tightly safeguarded as trade secrets by leading enterprises. Consequently, domestic running protective equipment often lacks high value-added core technologies, exhibits suboptimal service performance, and faces limited competitiveness in the consumer market. To address these challenges, it is crucial to analyze global development trends in running protective equipment technology, increase investments in scientific research, and overcome the technical barriers imposed by technologically advanced nations. Advancing the research and development of high-end running protective equipment is vital for fostering the growth of quality-driven sports industries and strengthening the nation's position in the global market.

Patents, as a critical embodiment of scientific and technological innovation, serve as a reliable indicator of emerging trends in scientific and technological progress [7]. According to estimates by the World Intellectual Property Organization (WIPO), the effective analysis and application of patent information can shorten the average research and development (R&D) cycle of enterprises by 60% and reduce scientific research expenditures by 40%. Consequently, in promoting the high-quality development of the sports industry, it is imperative for relevant authorities to uncover and capitalize on the substantial value embedded within patent data. In recent years, numerous scholars have investigated the technological evolution of sports-related products through patent analysis, providing competitive intelligence to support product R&D in sports manufacturing enterprises. For instance, Wang Jun (2013) conducted a comprehensive study of technological development trends in the football industry using patent data mining [8]. The analysis addressed multiple dimensions, including spatial and temporal distribution, patent types, and technical comparisons. The study identified key technical priorities in the field, such as materials used in football production, manufacturing processes, and auxiliary facilities. Similarly, Ming Yu et al. (2013) utilized patent analysis to conduct a comparative study of advancements in sports footwear technologies between Chinese sports enterprises and Nike [9]. Their findings indicated that Nike's input-output ratio in patent applications significantly exceeded that of Chinese companies, demonstrating superior integration, applicability, and innovation in R&D technologies. Yin Long et al. (2016) examined the development status of golf club technologies from a patent perspective [10]. The authors proposed that Chinese enterprises should increase their R&D investments in golf club technologies to establish competitive advantages for domestic brands. Additionally, Wan Yu et al. (2018) conducted a systematic analysis of the technological development trajectories, R&D priorities, and geographical distribution of international sports equipment patents [11]. Their findings revealed that the primary technological hotspots in sports equipment are concentrated in fitness apparatus, competitive gear, and training devices. Building on these foundational studies, the present research focuses on a targeted patent analysis of running protective equipment to elucidate technical development trends within this domain. This analysis aims to provide an evidence-based foundation for decision-making in technology R&D and patent strategies in China's running protective equipment industry, offering empirical insights into patent quality, technological trajectories, and commercialization potential.

## Analysis of basic patent information

### Data sources and search strategy

Patent data for this study were collected from the Patsnap Global Patent Database, and the retrieval was conducted in Jan 2025. The temporal coverage was set from 1898 to 2023. The starting year of 1898 was chosen because manual inspection confirmed that the earliest relevant patent related to running protective equipment dates back to that year. The endpoint of 2023 was selected in consideration of the typical 18-month delay between patent application and publication, which ensures the completeness and comparability of the data set. However, we acknowledge that the number of patents prior to the 1970s is extremely sparse and does not allow for meaningful statistical inference. To address this limitation, the data were treated in a differentiated manner: patents from 1898 to 1969 were included only as background information to illustrate the emergence and early experimentation of protective technologies, whereas the core analyses—such as technology evolution trends, core patent screening, and quality indicator evaluation—were based primarily on data from 1970 to 2023, when patenting activity expanded significantly alongside the industrialization of sports equipment and the maturation of patent systems. In addition, sparse early data were separately flagged and aggregated in trend figures to prevent them from distorting overall patterns. This approach preserves the completeness of the dataset while ensuring that the empirical results remain scientifically robust and analytically meaningful. The search strategy employed a combination of keywords (applied to the title, abstract, and claims fields) and IPC classifications, supplemented by manual screening to further improve relevance, which is widely recognized as a standard approach in patentometric research [12–14]. The keyword set was designed to directly reflect the research focus on running-related protective equipment. It included not only the core terms "protective running gear" and "running protective gear," but was also expanded to cover

semantically and technically relevant terms such as "running shoes," "sports footwear," "knee pad," "knee brace," "waist support," "waist protector," "sports protective clothing," and "compression wear." These terms were applied to the title, abstract, and claims fields, with Boolean operators and wildcards used to broaden coverage. To mitigate the limitations of keyword-based searching and reduce the risk of irrelevant retrievals, the search was constrained by International Patent Classification (IPC) codes associated with footwear, protective clothing, and sports equipment (e.g., A43B, A41D, A63B). High-frequency subclasses such as A43B5/06 (running shoes), A43B13/18 (elastic soles), A41D13/06 (knee or foot protectors), and A63B71/12 (body or leg protectors) were included. The selection of these subclasses was validated through iterative testing and manual verification to ensure their relevance to the domain of running protective equipment. This "keyword–classification" dual-filtering strategy follows the established guidelines in the OECD Patent Statistics Manual [12–13], and is consistent with prior empirical studies that have demonstrated its effectiveness in improving precision in patent searches [14]. To further enhance reliability, exclusion criteria were applied to filter out patents unrelated to running, such as industrial safety boots, mountaineering footwear, ski boots, and medical orthoses. Additionally, INPADOC family aggregation, applicant name harmonization, and manual review of abstracts and claims were performed during the cleaning phase. Beyond these procedures, we also recognize that equating patent volume directly with technological advancement may be misleading, since defensive patents, duplicate filings, and low-quality submissions could distort the measurement of innovation. To mitigate these risks, several safeguards were implemented. First, the INPADOC family aggregation and duplicate removal procedures minimized the inflationary effect of multiple filings for the same invention across jurisdictions. Second, manual screening excluded patents with marginal relevance to running protective equipment, thereby reducing the inclusion of defensive or strategically filed patents. Third, and most importantly, the analysis was complemented by quality-oriented indicators that move beyond simple counts, including forward citation frequency (as a proxy for technological impact and knowledge diffusion), patent family size (as an indicator of international value and strategic importance), number of claims (as a measure of technical breadth), and legal status such as survival period (as a reflection of technological durability). In this framework, patent counts are employed primarily as a proxy for the intensity of inventive activity, while the incorporation of quality-related measures ensures that the evaluation captures both the scale and the significance of technological innovation. This multi-dimensional approach enables a more rigorous and balanced interpretation of innovation dynamics in the field of running protective equipment. Together, these procedures ensured that the final data set was both comprehensive and reproducible, thereby providing a robust foundation for subsequent patentometric analysis. For the sake of clarity and readability, only the principal elements of the data collection strategy are presented here. The complete search queries, IPC class lists, exclusion criteria, and detailed data-cleaning procedures are provided in Appendices A–D.

**Analysis of global patent application trends**

Fig 1 presents the temporal trends in global and Chinese patent applications within the domain of running protective equipment, as well as the changing proportion of Chinese filings relative to global submissions. The data indicate that patent activity in this field dates back to 1898. From 1898 to 1970, technological advancement in this area was intermittent, with annual patent filings consistently remaining in the single digits. This period is therefore considered the nascent stage of innovation in running protective equipment, primarily marked by the foundational accumulation of technological knowledge. Between 1971 and 1999, rising living standards and growing public interest in fitness stimulated steady growth in patent activity. During this period, the annual number of applications increased from 11 to 62, marking a 5.63-fold rise [15]. This phase represents the formative stage of technological accumulation in running protective equipment.

In comparison to the global landscape, China's research and development efforts in the field of running protective equipment commenced at a relatively late stage, with the first patent applications filed in 1987. Since then, the trajectory of Chinese patent filings has remained broadly aligned with international trends. From the perspective of China's share in global patent activity, a clear upward trend has been observed on an annual basis. This growth has been supported

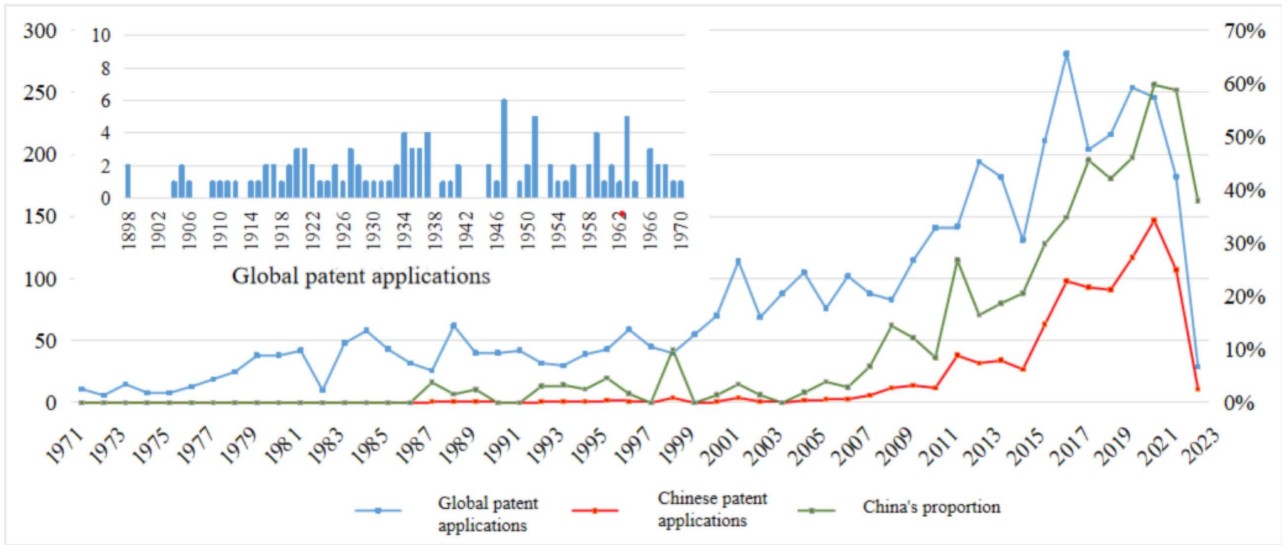

**Fig 1. Temporal trend of global and Chinese patent applications in the field of running protective equipment (1898–2023).** *Note*: Data were from Patsnap Global Patent Database (retrieval date: January 2025), constrained by IPC classes A43B, A41D, and A63B. Data were cleaned through INPA-DOC family aggregation, applicant name harmonization, and manual screening. The figure illustrates the temporal distribution of global and Chinese filings, showing a nascent stage (1898–1970), a technological accumulation phase (1971–1999), and rapid growth after 2000. China's share of global activity demonstrates a steady upward trajectory, reflecting policy support and enhanced innovation capacity.

by the implementation of national policy frameworks aimed at strengthening the sports industry and promoting mass fitness, reflecting China's strategy of innovation-driven development. This pattern reflects the continuous enhancement of innovation capacity and technological competitiveness among Chinese sports equipment manufacturers, particularly within the domain of running protective gear. These advancements are consistent with China's overarching strategy for innovation-driven development.

## Regional distribution of patent applications

The priority country of a patent application refers to the nation in which a specific technology, process, or invention is initially filed for intellectual property protection. In most cases, this country corresponds to the location where the underlying research and development activities are carried out. Accordingly, the designation of a priority country serves as a proxy for evaluating both the intensity of technological activity within a given nation and its contribution to the advancement of related scientific and technological innovation. Fig 2 presents the distribution of priority countries for patent applications in the domain of running protective equipment [16,17]. The data indicate that the United States—widely recognized as a global leader in both sports and technology—benefits from a vast consumer market and robust research and development capabilities. These advantages have translated into a dominant position in patent activity within this field, with the country holding 1,460 patents, accounting for 33.13% of the global total. China ranks second, with 929 patents, representing 21.08% of worldwide filings. As an emerging economy, China has rapidly developed its intellectual property regime since the establishment of its patent system in 1985. By 2021, it had maintained the highest volume of international patent applications for three consecutive years. In the specific field of running protective equipment, China has achieved significant breakthroughs and technological advancements, contributing to a marked increase in patent output. Other leading countries include Germany (324 patents, 7.35%), Switzerland (243, 5.14%), the United Kingdom (135, 3.06%), Canada (102, 2.31%), Italy (102, 2.31%), Japan (96, 2.18%), the Netherlands (75, 1.70%), and France (62, 1.40%). These figures

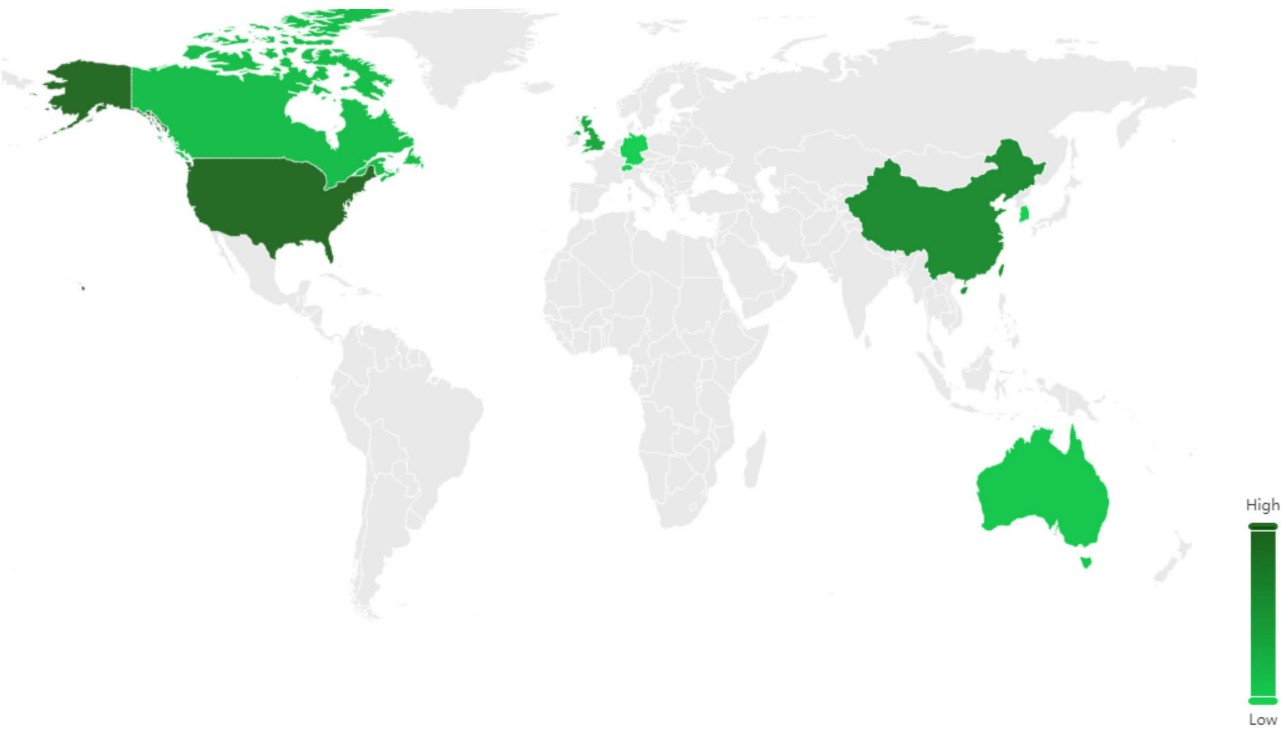

**Fig 2. Priority country distribution of patents related to running protective equipment.** *Note*: Data were from Patsnap Global Patent Database (retrieval date: January 2025). The figure shows the geographical distribution of priority filings, identifying the United States (33.13%) and China (21.08%) as leading contributors, followed by Germany, Switzerland, and the United Kingdom. Together, the top ten countries account for nearly 80% of global filings, underscoring the concentration of technological activity in a limited set of nations with advanced R&D and intellectual property systems.

represent between three and ten entities. From the perspective of national distribution, aside from China, the leading countries prioritizing patent applications in the field of running protective equipment are primarily developed Western nations. This pattern suggests that these countries maintain a technological advantage in this domain, supported by a long-standing accumulation of technical expertise and well-established intellectual property protection systems. Further analysis indicates that patent filings from the top ten priority countries account for approximately 79.66% of the global total [15]. This high level of concentration underscores the dominance of a limited number of technologically advanced nations in driving innovation within the field of running protective equipment.

**Global distribution of patent applicants**

Table 1 presents statistical data on the top ten high-output patent applicants in the field of running protective equipment. In terms of application volume, Nike, a cloud-based enterprise, and Adidas AG rank as the top three entities, with 95, 93, and 35 applications, respectively [15]. Notably, the top two positions are occupied by U.S.-based applicants, reflecting a substantial quantitative advantage. This underscores the United States' dominant position in this technological domain, characterized by robust research and development capabilities and considerable technical influence. From a geographical perspective, the distribution of patent applicants comprises three from the United States, two from Germany, and five from China. This distribution suggests that, under national strategies promoting intelligent manufacturing and technological advancement, China's running protective equipment sector has fostered a dynamic innovation ecosystem. Moreover, there is growing momentum among domestic innovators, as evidenced by increased engagement in

**Table 1. Relevant Information of Global Important Applicants for Running Protective Equipment.**

| sort | Patent right | global Quantity requested | Application share | Applications in the past five years | Last five years Take up a proportion of | nationality |
|---|---|---|---|---|---|---|
| 1 | Nike | 95 | 2.16% | 22 | 23.16% | USA |
| 2 | Cloud company | 93 | 2.11% | 17 | 18.28% | USA |
| 3 | Adidas | 35 | 0.79% | 2 | 5.71% | GER |
| 4 | Jiangyin Tianguang Technology | 22 | 0.50% | 10 | 45.46% | CHN |
| 5 | Anta | 17 | 0.39% | 16 | 94.12% | CHN |
| 6 | Fujian Hongxing Erke Sporting Goods | 17 | 0.39% | 8 | 47.06% | CHN |
| 7 | Frampton Ellis III | 12 | 0.27% | 0 | 0.00% | USA |
| 8 | Puma Europe | 12 | 0.27% | 1 | 8.33% | GER |
| 9 | Li Ning (China) Sporting Goods | 11 | 0.25% | 7 | 63.64% | CHN |
| 10 | Stange sports protection technology | 11 | 0.25% | 6 | 54.55% | CHN |

technological development and heightened awareness of intellectual property rights. However, when assessing the ratio of patent applications to the number of research and development entities, it becomes evident that although China has a larger number of R&D institutions, its patent output remains comparatively low. This discrepancy highlights a gap between innovation potential and the effective transformation of research outcomes into intellectual property assets. The disparity between Chinese entities and leading global corporations in the field of running protective equipment remains significant. This suggests that China currently lacks a dominant enterprise in this domain, thereby highlighting the urgent need to enhance the competitiveness of its innovation-driven actors. A further analysis of patent application shares over the past five years reveals that European and American firms—represented by Nike, a cloud-based company, and Adidas AG— have contributed a relatively small proportion of recent filings. This trend indicates that these well-established corporations have largely completed their foundational technological accumulation in this sector. Consequently, their current investment in research and development related to running protective equipment appears limited, reflecting a reduced level of innovation activity.

In contrast, Chinese companies—such as Tianguang Technology, Anta, and Li Ning—exhibited a relatively high proportion of patent applications during the same period. This indicates that these emerging market enterprises, despite their later entry into the sector, are actively engaged in a phase of technological catch-up. The combined influence of strong policy support and substantial market demand has significantly stimulated both innovation activity and institutional engagement. As a result, these firms have achieved a notable concentration of patent output over the past five years.

## IPC technical analysis

The International Patent Classification (IPC) system, established by the World Intellectual Property Organization, is based on defined technical criteria and is designed to categorize patents according to their technological content. The structure and distribution of IPC categories provide valuable insights into the technological orientation and developmental trajectory of a given field [18]. In this study, a total of 4,407 patents related to running protective equipment were collected and classified according to IPC codes for statistical analysis. This effort aims to provide a technical reference for the development of patent strategies by enterprises, research institutions, universities, and other innovation-oriented entities operating within this domain. As shown in Table 2, the top ten IPC patent classifications are primarily concentrated in three areas: running shoes, structural and material optimization, and body protection. Among these, patents related to running shoes constitute the largest share, with 1,895 filings, accounting for 42.99% of the overall patent landscape. This dominance is largely attributable to specific technical subclasses, including A43B5/06 (running shoes; track and field shoes), A43B13/18 (elastic soles), A43B13/12 (multi-layer soles composed of different materials), A43B5/00 (sports footwear equipped with

**Table 2. Top 10 IPC Classification Technologies in the World.**

| Ranking | IPC | Number of patents/pieces | Patents accounted for % | Technical meaning |
|---|---|---|---|---|
| 1 | A43B5/06 | 560 | 12.71% | Running shoes; Track shoes |
| 2 | A43B13/18 | 537 | 12.19% | Elastic sole |
| 3 | A43B13/14 | 445 | 10.10% | Characterized by structural shape |
| 4 | A43B13/12 | 325 | 7.38% | Multiple layers of soles of different materials |
| 5 | A43B5/00 | 311 | 7.06% | Sports footwear |
| 6 | A63B71/12 | 170 | 3.86% | For the body or legs |
| 7 | A41D13/05 | 167 | 3.79% | It's just a special part of the body |
| 8 | A43B13/20 | 162 | 3.68% | Inflatable sole |
| 9 | A41D13/06 | 161 | 3.65% | Knee pads or foot protectors |
| 10 | A43B13/04 | 157 | 3.56% | Plastic, rubber or vulcanized fiber |

non-slip devices, such as cleats), and A43B13/20 (inflatable soles). These findings indicate that, within the domain of running protective equipment, technologies focusing on sole-based shock absorption and anti-slip functionality represent a major area of technological concentration. The category pertaining to structural and material optimization encompasses 602 patents, representing 13.66% of total filings. This group is primarily characterized by technologies classified under A43B13/14 (defined by structural shape) and A43B13/04 (comprising plastic, rubber, or vulcanized fiber), along with other related subclasses. Additionally, the body protection category includes 498 patents, accounting for 11.3% of the total. This segment mainly comprises technologies classified under A63B71/12 (protective devices for the body or legs), A41D13/05 (garments designed for the protection of specific body parts), and A41D13/06 (knee pads and foot protection), along with other associated technical classifications [15].

From the perspective of IPC technology development trends, the three principal technical domains—running shoes, structural and material optimization, and body protection—have demonstrated a relatively consistent trajectory in patent application growth. Prior to 2000, these areas were in a phase of technological latency, characterized by sporadic filings and the absence of a discernible development trend. However, after 2000, they transitioned into a stage of technological maturity, marked by a significant acceleration in patent output. During this period, patent applications followed a generally upward yet fluctuating trajectory. Notably, a total of 1,878 applications were filed across the three domains, accounting for 42.61% of all patents submitted. This shift signifies the onset of a "golden age" for technological advancement in the field of running protective equipment. From a comparative perspective, both the cumulative volume and growth rate of patents related to running shoes significantly exceed those in structural and material optimization, as well as body protection [15]. This indicates that running shoes constitute a primary research focus within the broader technological landscape of running protective equipment. Furthermore, this trend indicates that technological competition among innovative actors is primarily concentrated in the domain of running shoes. The underlying rationale is that modern running shoes have become essential athletic equipment, embodying a sophisticated integration of engineering and technology. Their structural features—such as support, cushioning, and elasticity—play a critical role in mitigating ankle injuries among runners. Concurrently, the growing popularity of running, coupled with continuous advancements in shoe functionality, has significantly intensified consumer demand for high-performance footwear. In response to these market dynamics, numerous sports equipment manufacturers are actively investing in research and development to enhance running shoe technologies, increase market share, and maximize economic returns.

## Analysis of patent application types and validity

In view of the strong research and development momentum demonstrated by numerous enterprises in the field of running protective equipment—coupled with significant technological advancements—it is essential to examine both the types and

 

legal statuses of existing patents. Such analysis provides critical reference value for the formulation of effective patent strategies within the industry [19]. Fig 3 presents a comprehensive overview of patent classifications and their legal statuses related to running protective equipment. Among the various categories, invention patents dominate the landscape, with 3,402 applications accounting for 77.20% of the total. This is followed by design patents and utility models, with 512 and 493 applications respectively, representing 11.62% and 11.19% [15]. These figures indicate that invention patents serve as the primary form of intellectual property protection in this field, highlighting the significance of running protective equipment in safeguarding athletes' health and preventing running-related injuries. Moreover, the manufacturing of such equipment is subject to rigorous quality standards and technical specifications, thereby necessitating a high degree of innovation in R&D activities. Simultaneously, due to the geographic concentration of core technologies and the proprietary nature of enterprise-level expertise, it is imperative for R&D personnel to overcome patent barriers through sustained technological innovation. As a result, invention patents continue to represent the predominant form of protection within the patent landscape of this sector.

From the perspective of legal status, a total of 1,519 patents in the field of running protective equipment are currently authorized and valid, accounting for 34.47% of all filings. In contrast, 2,227 patents are classified as invalid, representing 50.53% of the total. These figures suggest that the overall quality of patents in this domain remains relatively low, indicating limited market applicability and weak technological durability. Further analysis reveals that unpaid annual fees account for 23.05% of the invalid patents, while patent withdrawals constitute 10.98% [15]. This highlights a significant imbalance between the intellectual property revenues generated by many patents in the sector and the costs required for their maintenance. Challenges related to commercialization and insufficient economic returns have contributed to the invalidation of a substantial number of patents due to non-payment of renewal fees or voluntary withdrawal. This high invalidation rate underscores the limitations of interpreting total patent counts as a straightforward measure of innovation capacity. Importantly, however, invalidation does not necessarily equate to a lack of innovation value: even patents that have lapsed or were withdrawn often reflected novel technical solutions at the time of filing, and thus provide meaningful evidence of inventive activity in their historical context. Accordingly, while Figs 1, 2 and 4 and Tables 1 and 2 present trend charts and structural distributions based on overall filing activity, they are primarily intended to capture the intensity and diffusion of inventive efforts rather than the long-term durability or substantive impact of technologies. To ensure robustness, the subsequent analyses complement volume-based measures with quality-oriented indicators—including forward citation

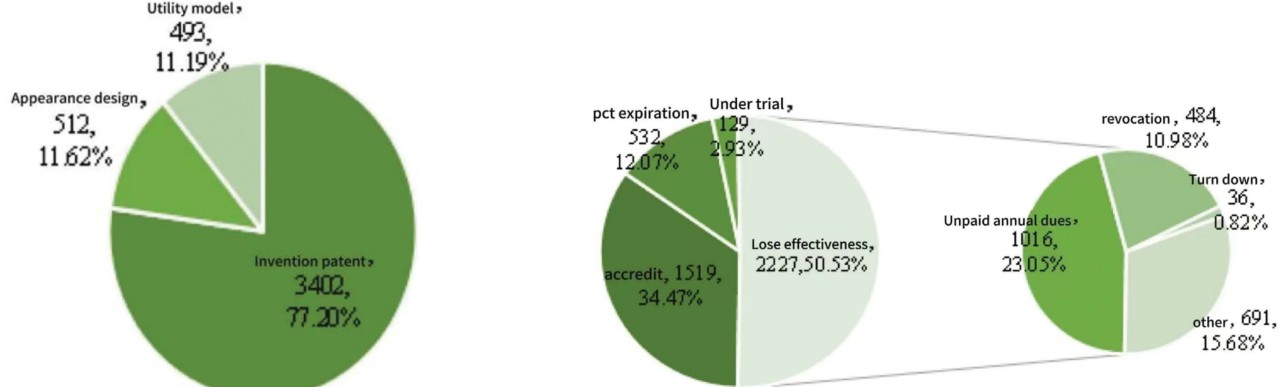

**Fig 3. Patent types and legal status in the field of running protective equipment.** *Note*: Data were from Patsnap Global Patent Database (retrieval date: January 2025). The figure shows the composition of invention, utility model, and design patents, alongside their legal statuses. Invention patents dominate (77.20%), but only 34.47% remain valid, while over half (50.53%) have lapsed, primarily due to unpaid annual fees or voluntary withdrawals. This reflects both the innovation focus of the sector and challenges in commercialization and long-term maintenance.

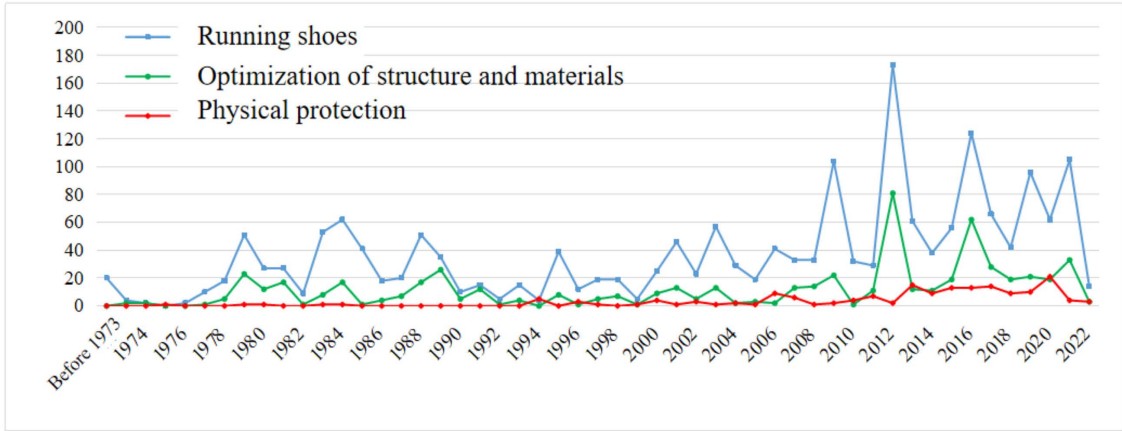

**Fig 4. Application trends of key IPC subclasses in the field of running protective equipment.** *Note*: Data were from Patsnap Global Patent Database (retrieval date: January 2025). IPC subclasses include A43B5/06 (running shoes), A43B13/18 (elastic soles), A43B13/12 (multi-layer soles), and A41D13/06 (knee or foot protectors). The figure highlights the dominance of running shoe technologies, particularly those related to sole structures and cushioning systems, as well as the steady growth of protective clothing and knee pad technologies after 2000.

frequency, patent family size, number of claims, and survival period—as well as the TOPSIS-based identification of core patents. This integrated approach balances scale with quality, enabling a more rigorous and nuanced interpretation of innovation dynamics in the field of running protective equipment.

## Analysis of patent technology evolution

### Core patent screening

**Core patent screening indicators.** As a fundamental aspect of patent analysis, the identification and extraction of core patents are essential for uncovering key technologies within the target domain. This process plays a pivotal role in elucidating the technological evolution of the field [20]. Drawing upon the methodological frameworks established by Li Meining [21], Xu Xin [22], and Wang Zhixing [23], this study develops a core patent evaluation index system encompassing the dimensions of technological advancement, application potential, and operational stability, as illustrated in Table 3. In determining the indicator weights, this study adopts a combined weighting approach that integrates both subjective and objective factors to ensure scientific rigor and practical validity. The subjective factors are derived from expert evaluations of the relative importance of the three dimensions—technological advancement, application prospects,

**Table 3. Core patent screening index system.**

| Primary index | Secondary index | weight |
|---|---|---|
| The advanced nature of patented technology | Number of patent citations | 0.11 |
| | Cited frequency | 0.09 |
| | Number of patentees | 0.18 |
| Patent application prospect | Number of IPC class numbers | 0.17 |
| | Quantity of claim | 0.09 |
| | Number of patents in the same family | 0.14 |
| Patent operation stability | Number of pledges, transfers and licenses | 0.07 |
| | Survival time | 0.08 |
| | Manual pages | 0.07 |

and operational stability. The experts included university scholars specializing in intellectual property and innovation management, technical professionals with practical experience in the field of sports equipment, and patent analysts with more than two years of experience in patent evaluation. Their professional backgrounds ensure that the subjective judgments incorporate theoretical, practical, and analytical perspectives, thereby compensating for potential biases that may arise when relying solely on quantitative data. The objective factors are based on measurable indicators obtained from the patent database and include: (1) in the dimension of technological advancement, the number of cited patent documents, forward citation frequency, and the number of patents; (2) in the dimension of application prospects, the number of IPC classifications, the number of claims, and patent family size; and (3) in the dimension of operational stability, the number of pledges/transfers/licenses, the survival period, and the number of specification pages. The objective weights were calculated using the entropy method to capture the informational contribution of each indicator. The final weights were obtained by integrating the subjective evaluations with the objective results, and these combined weights were subsequently applied within the TOPSIS model to calculate the comprehensive scores of patents in the field of running protective equipment. This process ensures that the weighting system is both data-driven and enriched by multi-perspective expert knowledge, thereby enhancing the accuracy and robustness of high-quality patent identification.

To reconstruct the technological evolution pathways, we adopted a multi-step approach combining quantitative screening with qualitative validation. First, a keyword–IPC dual filtering strategy was applied to ensure that the dataset captured both semantic relevance and technical specificity. Second, the TOPSIS evaluation framework was employed to identify representative core patents. Third, these core patents were validated through manual review by independent coders with expertise in patent analysis and sports equipment technology, ensuring interpretive reliability. Finally, by analyzing the temporal distribution and technological clustering of the validated patents, developmental trajectories were synthesized into visual maps (Figs 5–8). This integrated quantitative–qualitative procedure provides methodological rigor and interpretability, ensuring that the presented trajectories are both empirically grounded and practically meaningful. It should be noted that all company-related cases presented in this study (e.g., Nike, Li Ning, Waco, Puma, Brooks, etc.) are included solely as illustrative examples of patenting activity, serving to highlight representative technological features or trends rather than to endorse or promote any specific company or product.

## Analysis of core patent technology

As illustrated in Fig 5, the technological evolution within the domain of running protective equipment primarily centers on three key components: running shoes, sports protective apparel, and knee pads. These three domains not only reflect product-level distinctions but also align with the International Patent Classification (IPC) system, with running shoes corresponding mainly to A43B (footwear), sports protective apparel to A41D (protective clothing), and knee pads to A63B (sporting equipment and protective devices). The observed evolution encompasses a range of technological innovations. For running shoes (A43B), relevant subclasses include A43B13/18 (sole stabilization plates and elastic soles), A43B13/12 (compressible midsoles), A43B13/26 (shock-absorbing soles), and A43B23/02 (structural support systems for the shoe body). For sports protective apparel (A41D), innovations correspond to A41D31/02 (optimization of textile materials), A41D1/02 (stitching and garment construction techniques), and A41D13/05–06 (integration of protective elements into garments, such as knee or foot protectors). For knee pads (A63B), the relevant subclass is A63B71/12, which covers protective devices for the body or legs, including structural design improvements, joint fabrication methods, fitting and fastening mechanisms, and optimizations of shock-absorbing devices [15].

**Technical evolution analysis of running shoes.** As illustrated in Fig 6, the evolution of patented technologies for running shoes primarily centers on three key areas: stability mechanisms, sole support and protection structures, and the diversification of safety functionalities. These trajectories correspond mainly to IPC subclass A43B (footwear). Within this subclass, the observed technological innovations map onto specific categories: sole stabilization plates to A43B13/18 (elastic soles and stabilization structures), compressible midsole materials to A43B13/12 (layered or compressible soles),

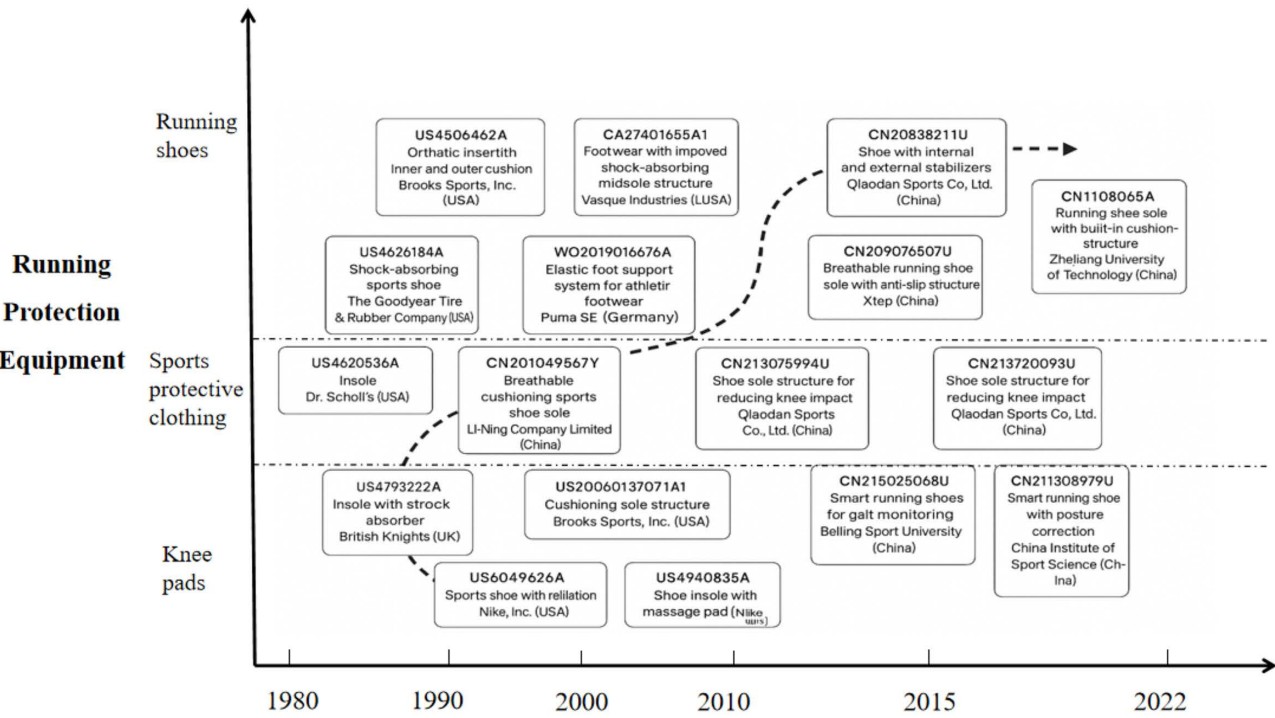

**Fig 5. Technological evolution pathways of core patents in running protective equipment.** *Note*: Data were from Patsnap Global Patent Database (retrieval date: January 2025). The diagram is based on 19 core patents identified through TOPSIS evaluation (forward citations, family size, claims, survival period) and validated by manual review. It outlines three subdomains—running shoes, protective apparel, and knee pads—highlighting key milestones in material optimization, structural design, and functional integration.

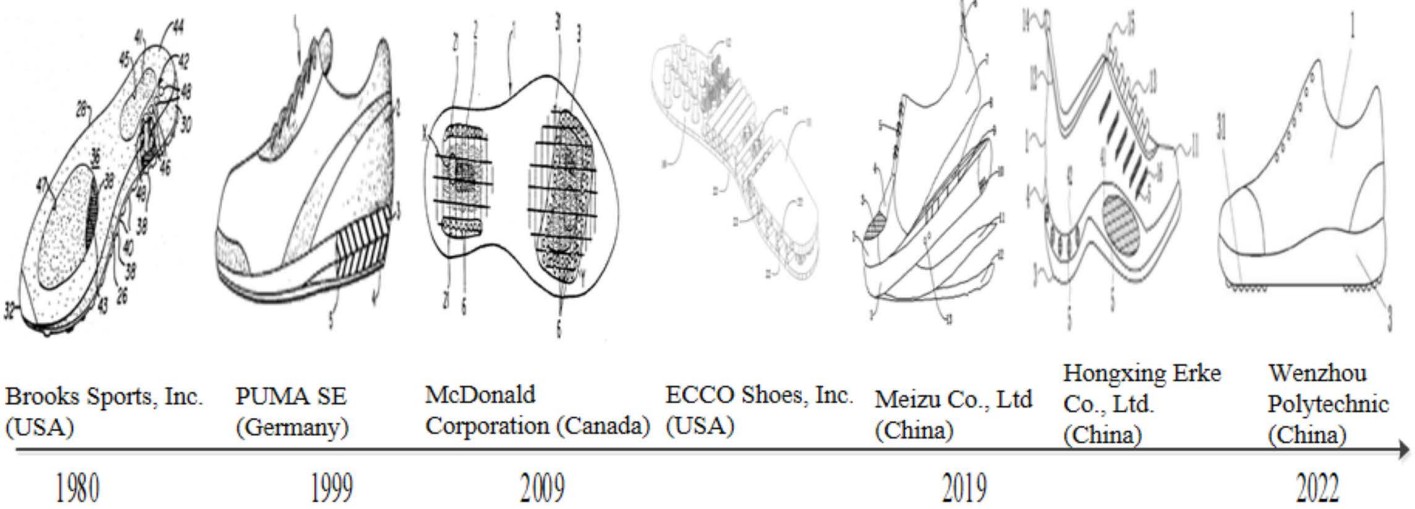

**Fig 6. Evolution of patented technologies in running shoes.** *Note:* Data were from Patsnap Global Patent Database (retrieval date: January 2025). Representative patents were selected through TOPSIS-based screening and manual review, covering stability mechanisms, multi-layer cushioning, lightweight material integration, ventilation systems, and antibacterial protection. Example patents include contributions from U.S. (e.g., Brooks), German (e.g., Puma), and Chinese (e.g., Hongxing Erke) enterprises, reflecting the evolution of durability, comfort, and safety functions in running shoes.

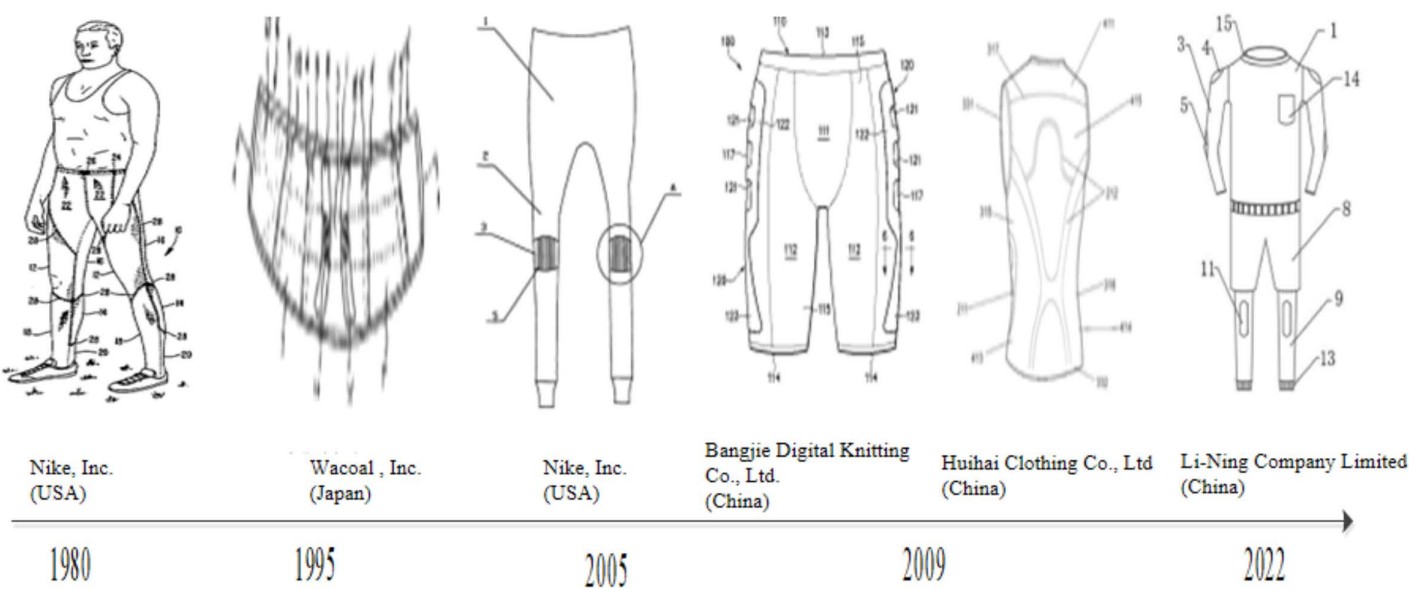

**Fig 7. Evolution of patented technologies in sports protective clothing.** *Note:* Data were from Patsnap Global Patent Database (retrieval date: January 2025). Patents selected via TOPSIS and manual review illustrate innovations in thermal regulation, abrasion resistance, waist and knee support, muscle compression, and modular protective pad integration. Example patents include those from U.S. (Nike), Japan (Waco), and China (Li Ning, Yangzhou Huihai Clothing). The figure demonstrates the transition of protective apparel from single-function garments to multi-functional protective systems.

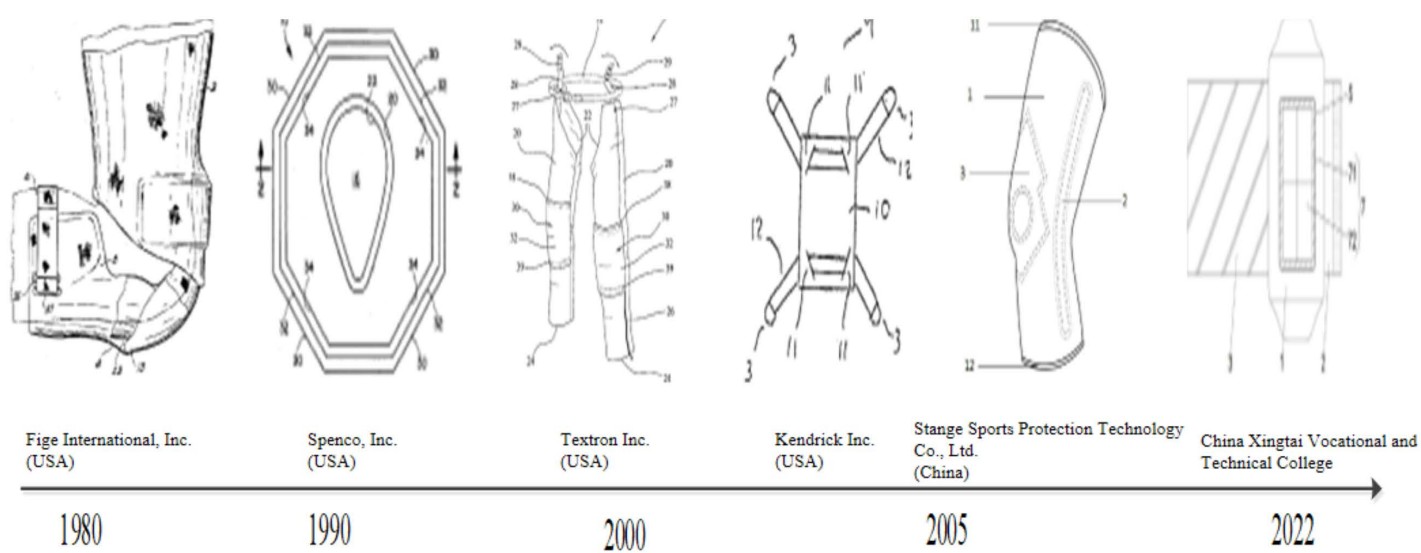

**Fig 8. Evolution of patented technologies in knee braces.** *Note:* Data were from Patsnap Global Patent Database (retrieval date: January 2025). Representative patents identified through TOPSIS and manual review highlight improvements in abrasion resistance, anatomical fit, fastening mechanisms, cushioning systems, and therapeutic functions. Example patents include those from U.S. (Fige International, Spenco), Textron, and China (Stange Sports Protection). The figure reflects the trend toward integrating preventive and rehabilitative roles in knee pad technologies.

structural support systems for the shoe body to A43B23/02 (reinforcements), elastic components and internal shock absorption systems to A43B13/26 (shock-absorbing soles), dehumidification and ventilation mechanisms to A43B7/14 (ventilation devices), antibacterial improvements to A43B17/00 (hygienic footwear), illumination systems to A43B5/06 (special footwear with integrated lighting), and material innovations such as PU, EVA, and NBR composites to A43B13/20 (polymeric or composite sole materials) [15].

Due to the forward-leaning posture of runners and the inherent instability of the sole during motion, excessive pronation frequently occurs in the knees and feet. If left uncorrected, this condition may lead to foot bone deformities, knee varus, osteoarthritis, and other related musculoskeletal disorders. In response, early innovations in running shoe design primarily focused on enhancing sole stability mechanisms to regulate the rotational movement of the foot and knee.

Patents from US firms (e.g., Brooks) on dual-layer cushioning structures that combine elastic components of varying hardness, aimed at improving shock absorption and biomechanical stability. The dual-layer cushioning design of this stability device absorbs the mechanical stresses generated during running. This innovation mitigates impact forces between the foot and the shoe body, enabling the footwear to better withstand lateral forces experienced by the lower limbs. Building upon this concept, Patents from Germany firms (e.g., Puma) emphasize midsole cushioning materials and structural inserts designed to mitigate lateral displacement and regulate pronation.

With improvements in the fundamental performance characteristics of running shoes, efforts to enhance durability, comfort, and protective functionality have increasingly focused on optimizing sole support and protection systems. During running, the impact forces and pressure generated by lower limb movement are primary factors contributing to both footwear degradation and injury risk. In response to these challenges, Patents from Canada (e.g., Mcdonald) have developed a multi-structure support system for footwear, based on the principle of fluid transfer. This system incorporates multiple elastic structural components, micro air cushions, and compression elements embedded with buoyancy magnets. As a result, the shoe exhibits compression resistance and shock absorption performance. Under this system, the impact forces and kinetic energy generated during running can be absorbed and dissipated, thereby reducing injury risk while enhancing the structural durability of the shoe. However, as advancements in basic performance characteristics continue, the integration of numerous functional elements has led to a substantial increase in shoe weight, along with considerable spatial compression, thereby limiting foot mobility. As a result, this not only reduces flexibility and comfort but also increases the risk of injuries to various parts of the foot, including the toes, arches, and heels. To address this issue, Patents from US (e.g., Aibu Shoes Company) conducted a comprehensive study on the physiological structure of the human foot. Guided by the principles of product simplification and functional optimization, the company adopted an integrated and lightweight shoe design. By incorporating a combination of specialized low-density PU variants (with a density of 0.35 g/cm³), EVA polymers, and NBR rubber materials, the design balances the essential structural performance of the shoe body with enhanced comfort and flexibility.

Driven by rapid advancements in materials science, intelligent equipment, and manufacturing technologies, the safety performance of running shoes has undergone comprehensive enhancement and expansion. Influenced by factors such as extended working hours, irregular rest patterns, and the pursuit of exercise efficiency, there is a growing trend among runners favoring evening workouts. In response to this development, some Chinese patents explore fiber-optic illumination in running shoes to enhance visibility and safety during nighttime exercise. Furthermore, to tackle persistent challenges related to poor breathability and inadequate moisture management in existing running shoes, Chinese patents include innovations in upper materials and active ventilation systems to improve breathability and moisture management. The active ventilation system primarily consists of a temperature and humidity sensor, micro air pump, airbag, inlet and outlet tubes, ventilation apertures, and one-way intake and exhaust valves. It is designed to automatically monitor the internal temperature and humidity of the shoe. When these parameters exceed predefined thresholds, the system actively expels heat and moisture, thereby maintaining a consistently dry and comfortable internal environment. To address the risk of bacterial infections resulting from prolonged physical activity and to safeguard foot health, Wenzhou Vocational

and Technical College (China) has implemented a targeted strategy to regulate internal humidity and suppress microbial growth within footwear. This approach involves integrating antibacterial and dehumidifying interlayers into the sole. The composite structure primarily comprises bactericidal agents and desiccants, housed within an internal mounting slot that allows for convenient replacement.

**Evolution analysis of sports protective clothing technology.** Fig 7 illustrates the evolution of patented technologies in the domain of sportswear. Within the field of sports clothing technology, the development of key patents has primarily focused on several critical areas, including thermal regulation and insulation, abrasion resistance, knee and waist protection, muscle strain mitigation, and shock absorption. These domains correspond mainly to IPC subclass A41D (protective clothing). Within this subclass, the observed technological innovations map onto specific categories: thermal regulation and insulation align with A41D31/02 (temperature-regulating clothing), abrasion resistance with A41D1/02 (improvements in fabrics and stitching), knee and waist protection with A41D13/05–06 (garments protecting specific body parts), compression and strain mitigation with A41D31/04 (compression or performance-enhancing apparel), and integration of protective pads with extended applications of A41D13/05 (modular protective apparel) [15].

The initial advancements in sportswear technology primarily focused on enhancing the thermal regulation and abrasion resistance properties of garments. It is well established that runners generate substantial heat during exercise. Given the temperature sensitivity of muscles in an active state and the variability of environmental conditions, maintaining a stable body temperature is essential. This thermal stability helps prevent muscle spasms, stiffness, and other symptoms affecting muscles, tendons, and surrounding soft tissues that may result from exposure to excessively cold or hot environments. To address this issue, patents from US firms such Nike have developed a line of sports protective clothing. The outer layer of this apparel is made from a specialized mesh material designed to facilitate heat dissipation. Simultaneously, a thermostatic component is integrated into muscle-sensitive areas using four-needle flat-seam technology. This component comprises lightweight composite materials—such as wool and knitted spandex—that offer ventilation and thermal insulation. To reduce friction-related injuries that runners may sustain during falls, patents from Japanese firms (e.g., Waco) emphasize abrasion resistance and comfort, illustrating the technical trajectory of protective sportswear.

With the continuous advancement of sports protective clothing, addressing issues such as knee injuries and lumbar strain—common occurrences during running—has become a primary focus. Consequently, research and development in this field increasingly emphasize enhancing protective functionality for both the knees and waist. While maintaining comfort and a lightweight design as fundamental principles, US company have developed a type of waistband-integrated sportswear made from stretchable woven fabric. This garment features a raised, contoured fit that conforms to the abdomen and waist. In addition, it incorporates elastic resin-based bone braces positioned along both sides of the waist and spine. These braces are specifically engineered to expand in response to changes in abdominal pressure during physical activity, thereby stabilizing the lumbar spine and alleviating fatigue or pain in the lower back. To simultaneously protect the knees while maintaining the comfort typically associated with traditional sweatpants, some Chinese patents developed a pair of sweatpants that integrates both comfort and knee-support functionality. The pants incorporate a knee brace component featuring a groove anatomically tailored to fit the outer contour of the human knee. This design minimizes friction between the knee brace and the lower limb, reducing discomfort caused by close contact between the fabric and the runner's skin. As a result, it prevents potential knee injuries and improves overall wearing comfort.

Amid growing concerns over sports-related injuries and heightened public awareness of health, consumer expectations for sportswear have evolved beyond the localized protection of areas such as the knees and lumbar spine. These expectations now encompass comprehensive protection that targets multiple regions of the human body. In response, the Chinese sportswear companies have introduced an innovative reinforced bodysuit. This form-fitting garment employs a four-stitch, six-thread sewing technique to integrate hollow, breathable mesh fabric—composed of spandex and nylon—into the lower back and outer thigh areas. By applying compression principles, the design incorporates laser-cut laminated fabrics and thermoplastic (TP) adhesive cloth, which are strategically positioned over key muscle regions, including the

gluteal area, lateral thigh muscles, abdominal muscles, and back muscles. This design reduces the risk of strain in critical muscle groups commonly engaged during running, such as the rectus abdominis, trapezius, gluteus maximus, semi-membranosus, and semitendinosus. To minimize the costs associated with acquiring multiple pieces of protective gear and to diversify functional coverage, patents from chines firms have developed an integrated solution based on modular connection principles. This approach employs mechanical, detachable, and electrical connection methods to embed shock-absorbing silicone pads into garments. These pads are strategically positioned in high-impact areas, including the shoulders, elbows, back, and knees, providing simultaneous protection to multiple vulnerable body parts. This innovation addresses the challenge faced by runners who would otherwise need to purchase separate protective equipment for each individual region.

**Evolution analysis of knee brace technology.** Fig 8 illustrates the evolution of patented technology in the domain of knee pads. The progression of key patents within this field primarily focuses on three core aspects: abrasion and slip resistance, ergonomic comfort, and the enhancement of protective performance. These trajectories correspond mainly to IPC subclass A63B71/12 (protective devices for the body or legs). Within this subclass, the observed technological innovations map onto specific categories: abrasion- and slip-resistant surface treatments to A63B71/12 (abrasion- and slip-resistant protectors), ergonomic comfort designs such as anatomically contoured structures to A63B71/12 (ergonomic protective devices), joint articulation techniques to A63B71/12 (articulated joints for body protectors), fastening and stabilization mechanisms to A63B71/12 (securing and attachment systems), cushioning optimization with viscoelastic foams or elastomeric inserts to A63B71/12 (shock-absorbing protective devices), and the development of impact-resistant materials to A63B71/12 (composite or reinforced materials for high-impact protection) [15].

In the initial applications of knee pads, functional limitations were evident due to coarse surface friction treatments and inadequate fixation mechanisms, which often led to slippage, displacement, and insufficient shock absorption during use. To address these shortcomings, Fige International Company (United States) developed an innovative knee brace with a double-layer overlapping composite structure. The brace incorporates a contoured concavity in the upper region that accommodates the patella and provides lateral support, thereby improving joint stability and reducing displacement. The depth of this concavity is restricted to no more than approximately 65% of the overall pad thickness, ensuring that the cushioning layer delivers shock absorption without impairing the flexibility and mobility of the knee during exercise. From a materials perspective, the upper cushioning section employs viscoelastic polyurethane foam or similar foamed polymers, with a storage modulus controlled within the range of 10–45 psi/cycle. This configuration enables the pad to absorb vertical impact forces while minimizing rebound effects transmitted back to the joint. The surface in direct contact with the knee is composed of a soft and breathable fabric, which enhances user comfort and prevents material migration that might otherwise stain the skin or clothing. The lower region of the pad is made of a more rigid yet flexible material such as ethylene vinyl acetate (EVA), which maintains the structural stability of the device while providing frictional resistance against the supporting surface without adhering to it. Through the integration of these structural and material specifications, the knee brace overcomes the shortcomings of traditional designs—namely instability, slippage, and inadequate shock absorption—while improving both stability and comfort for users during running and other forms of vigorous exercise. Subsequently, some us companies introduced an advanced knee pad aimed at further improving stability through refinement of the existing design. Retaining the double-layer structure, the new knee pad was optimized based on the physiological characteristics of the human knee. In particular, modifications to the external contour enhance anatomical conformity, resulting in a more precise fit and increased stability during use.

With the enhancement of the fundamental functions of knee pads, there is increasing demand for improved wearing comfort and the mitigation of issues such as poor blood circulation and skin abrasion caused by excessive compression. In response to this need, Textron has introduced an innovative type of knee pad. The primary structure consists of flexible materials that conform to human leg anatomy, including nylon mesh, polyester film plastic, and leather. Departing from traditional fixation methods that rely on cloth straps wrapped around the legs, the new design incorporates openings at

both the upper and lower sections of the knee pad. Flexible carabiner rings are integrated into these openings, enabling a secure fit without the need for tightly fastened straps. This approach alleviates the discomfort experienced by runners due to excessive pressure on the knees and legs caused by conventional strapping techniques. Furthermore, to enhance wearing comfort even further, some US companies have developed a flexible bow pad integrated into the knee pad. This component is constructed from neoprene and plastic foam fabric, offering improved tactile performance and anatomical conformity. Additionally, its distinctive attachment method—using a specially designed elastic woven belt—ensures that the pad maintains alignment with the runner's movements. As a result, this design not only enhances comfort but also addresses displacement and slippage, which are common problems associated with traditional knee braces.

With the rapid increase in the number of runners and the diversity in their physical fitness, running form, and training volume, knee injuries of varying severity have become increasingly common. This trend has prompted a heightened focus on improving the protective functions of knee pads in technical development. In response to this demand, some Chinese companies have designed an innovative sports knee guard. This product primarily consists of three components: a cylindrical sleeve woven from elastic fibers, a support strip constructed from flat spring material, and a massage pad made of soft silicone. These elements are integrated using advanced hot-press cloth bonding technology. During use, the double-fiber cylindrical sleeve provides enhanced compressive support, improving adherence between the knee brace and the knee joint. The support strip, composed of flat spring material, generates elastic resistance during deep knee flexion, offering cushioning performance and reducing the risk of strain. Furthermore, the triangular soft silicone massage pad delivers therapeutic stimulation to support quadriceps muscle health and acts as a buffer against collisions and impacts at the knee joint, thereby protecting critical anatomical structures such as the patella. Through the synergistic integration of these three elements, the protective efficacy of the knee brace has been enhanced, combining both preventive and rehabilitative functions.

## Conclusions and countermeasures

### Conclusion

Based on data from the Patsnap Global Patent Database, this study employed a patent bibliometric methodology to analyze innovation dynamics in running protective equipment. By examining application trends, geographical distribution, applicant profiles, IPC technology classifications, patent types, legal status, and technological evolution pathways, several key findings emerge.

(1) Temporal evolution of patenting activity demonstrates clear stages of development. The period from 1898 to 1970 reflects an embryonic phase with sporadic and low-volume filings, representing the initial emergence of protective technologies. From 1971 to 1999, technological accumulation was stimulated by rising public participation in running events and broader fitness movements, which generated sustained demand for protective equipment. After 2000, the field entered a phase of rapid expansion, driven by advances in material sciences, electronics, and composites, as well as heightened awareness of intellectual property protection. This staged progression, when mapped through IPC subclasses, indicates that innovation has consistently clustered in three domains—A43B (footwear), A41D (protective clothing), and A63B (sporting equipment and protective devices)—providing a stable framework for interpreting long-term trajectories.

(2) Geographical and applicant distribution patterns reveal significant asymmetries. Patent activity is concentrated in the United States, China, and Germany, with Western economies maintaining a sustained leadership position. The United States in particular exhibits a dual advantage in both volume and quality, reflected in higher forward citations, larger family sizes, and longer survival periods. Although China entered the field later, its patent output has grown steadily, reflecting rising innovation capacity. Nevertheless, the concentration of high-quality patents in subclasses such as A43B5/06 (specialized footwear) and A41D13/06 (protective apparel for knees and feet) remains dominated by firms in

advanced economies, underscoring the persistent gap between China's filing activity and its substantive technological influence.

(3) Patent type and legal status analysis highlights systemic challenges. Invention patents dominate the landscape, accounting for over three-quarters of all filings. However, only 34.47% of patents remain valid, while more than half have lapsed, frequently due to non-payment of annual fees. This imbalance reflects structural weaknesses in commercialization incentives, particularly evident in footwear-related subclasses (A43B), where lapse rates exceed half of all filings. Short survival periods and weak revenue streams indicate that enterprises often prioritize filing numbers over durable technological value, limiting the economic viability of innovations.

(4) Technological focus and evolutionary trajectories converge on three major domains: running shoes, sports protective apparel, and knee pads. These domains correspond respectively to IPC subclasses A43B, A41D, and A63B, anchoring product-level distinctions in the international classification framework. In footwear (A43B), R&D emphasizes stability mechanisms, midsole cushioning systems, and integrated safety functions, with technological hotspots including stabilization plates (A43B13/18), compressible midsoles (A43B13/12), and structural reinforcements (A43B23/02). In sports apparel (A41D), innovation focuses on thermal regulation (A41D31/02), abrasion resistance (A41D1/02), waist and knee support (A41D13/05–06), and multifunctional integration, reflecting a transition toward compression garments and modular protective designs. In knee pads (A63B71/12), research addresses abrasion resistance, ergonomic comfort, fastening and stabilization systems, cushioning optimization, and impact-resistant materials, illustrating the dual shift from preventive protection to rehabilitative functions. These IPC-grounded trajectories demonstrate a broader trend toward multifunctionality, intelligent integration, and performance optimization, confirming that the sector is evolving beyond incremental improvements into systemic innovation.

## Suggestions

Based on the aforementioned research findings, this paper aims to promote the development of China's running protective equipment industry by fostering technological innovation and enhancing strategic patent deployment among enterprises in the sector. Accordingly, the following countermeasures and policy recommendations are proposed:

(1) The delayed establishment of an effective intellectual property (IP) protection system in China, coupled with limited administrative experience, has resulted in several long-standing issues. These include an underdeveloped legal framework, inconsistent enforcement mechanisms, inadequate enterprise capacity to resolve IP disputes, and the generally low quality of intellectual property assets. Such challenges hinder not only the high-quality development of China's protective equipment industry but also significantly restrict enterprise-led research, development, and innovation. The legal status analysis provides direct evidence for this problem: only 34.47% of patents in the dataset are currently authorized and valid, while more than half have lapsed, a pattern that indicates insufficient incentives for long-term maintenance. When combined with the short average survival period of patents, this finding suggests that enterprises often prioritize filing numbers over durable value. This phenomenon, if left unaddressed, risks exacerbating the problem of "quantity over quality" in China's patent system. Notably, this pattern is especially pronounced in A43B subclasses (footwear patents), where lapse rates exceed half of all filings, underscoring that structural weaknesses are concentrated in specific IPC domains rather than uniformly distributed across the sector. To address these issues, it is necessary for the government to first reinforce the legal foundation of IP protection by promoting coordinated amendments to the Patent Law, Trademark Law, and Copyright Law. This effort should also involve standardizing procedures for IP litigation and increasing penalties for infringement and non-compliance. Second, the patent examination process and the broader intellectual property evaluation system must be further improved. While protecting the legitimate rights of innovators, the evaluation process should incorporate quality-based criteria, such as forward citation counts, family

size, and claims breadth, which our TOPSIS results have demonstrated to be reliable indicators of patent durability and market potential. Such quality-oriented governance not only reduces the risk of over-patenting but also encourages enterprises to focus on technologies with genuine commercial and technological value. Finally, a national early warning and emergency response mechanism for managing IP risks should be established. This is particularly important given our finding that high-value patents are concentrated in a small number of applicants: early identification of risks associated with these assets is crucial for securing China's competitive position. Collectively, these measures are expected to strengthen the international competitiveness of Chinese enterprises and provide more reliable institutional safeguards for sustained innovation.

(2) Fostering innovative leading enterprises is essential for enhancing the competitiveness and technological influence of China's protective equipment industry. Our applicant analysis shows that while China contributes the largest number of filings overall, patents with higher forward citations and longer maintenance periods are predominantly held by firms from advanced economies such as the United States, Germany, and Japan. For instance, in footwear-related subclasses (A43B5/06, A43B13/18), high-quality patents are clustered around a few global companies, whereas most Chinese enterprises exhibit low citation counts and limited international patent families. This asymmetry indicates that China's patent activity has not yet translated into equivalent technological influence. Enterprises serve as a dynamic force in driving innovation, with leading firms functioning as the cornerstone of scientific and technological advancement. These entities have the capacity to aggregate and integrate critical innovation resources—including talent, capital, and technology—while serving as service platforms, incubators, and accelerators through their demonstration and multiplier effects. Accordingly, targeted policy support should prioritize domestic enterprises active in IPC subclasses where China shows rising but fragile presence—such as A43B5/06 (special-function footwear) and A41D13/06 (protective apparel for knees and feet)—to ensure that emerging technological strengths translate into durable competitive advantages. On the one hand, relevant administrative departments must implement preferential policies such as tax deductions for R&D expenditures and accelerated depreciation of research equipment. These measures aim to unlock the innovation potential of enterprises and promote sustainable development by alleviating financial burdens. On the other hand, government procurement mechanisms should be leveraged to create demand-driven markets. Evidence from our dataset suggests that areas with active legal events (e.g., assignment, licensing) already demonstrate higher commercialization potential. Therefore, procurement programs should prioritize patents and technologies with documented transfer or licensing activity, which can accelerate their transformation from intellectual property assets into industrial outputs. Additionally, financing channels must be optimized to reduce operating costs and enhance innovation efficiency, with priority access granted to firms whose patents have passed our TOPSIS quality screening. Finally, the government should establish a streamlined mechanism for attracting high-level talent to the sector, particularly in firms located in the identified technological hotspots. Such measures can effectively bridge the gap between patenting activity and substantive innovation performance.

(3) Given current trends in the evolution of patents related to running protective equipment, it is imperative for Chinese enterprises to strengthen their awareness of intellectual property (IP) protection and adopt strategically aligned patent management practices. Our analysis of technological evolution clearly shows that innovation has shifted from traditional structural stabilization functions—such as sole support and cushioning—to more advanced materials (e.g., composites, breathable fabrics) and intelligent features (e.g., fiber-optic illumination, active ventilation). This trajectory demonstrates that future competitiveness will increasingly depend on capturing emerging technologies rather than solely reinforcing traditional designs. The formulation of effective patent strategies must integrate internal innovation capacity with external environmental dynamics to support sustainable industrial growth. Our IPC trend analysis indicates that subclasses A43B (footwear), A41D (protective clothing), and A63B (sporting equipment) remain dominant; however, within these classes, the fastest growth and highest citation intensities are found in subclasses

like A43B5/06 (running shoes with specialized functions) and A41D13/06 (knee or foot protectors). These hotspots highlight the technological frontiers where China has begun to establish a presence, and aligning R&D investment with these IPC subclasses can maximize both technological and commercial impact. In the early stages of technological development, enterprises should prioritize the accumulation of foundational technologies through a combination of approaches, including technology licensing, acquisition of high-value patents, and targeted talent recruitment. Meanwhile, firms should aim to integrate such core patents into their portfolios, either through direct R&D or collaborative arrangements. Concurrently, R&D investments should be significantly increased in critical segments such as running shoes, sports apparel, knee pads, and adjacent technologies to ensure precise alignment with emerging technological frontiers. Moreover, it is essential to proactively address technical barriers and mitigate the risk of patent infringement, particularly in response to challenges posed by foreign competitors. Evidence from our dataset shows that foreign applicants tend to maintain longer-lasting and higher-quality patents, which suggests that strategic foresight and diversification are necessary for Chinese enterprises to secure freedom-to-operate in contested technological spaces. Ultimately, strategic patent management should not be volume-driven but guided by the quality metrics we developed through TOPSIS screening. By focusing on patents with higher scores in forward citations, family size, and survival period, enterprises can ensure that their portfolios are resilient, defensible, and aligned with future industry trends.

(4) Currently, the field of running protection equipment is characterized by a high rate of patent invalidation and short maintenance periods. Our legal status analysis indicates that only one-third of patents remain valid, and the average survival time is short, suggesting a structural gap between innovation outputs and their commercial utilization. However, attributing this gap solely to commercialization failure is insufficient. Instead, the evidence suggests that limited transfer mechanisms, weak incentives for maintenance, and the prevalence of defensive or redundant filings are primary contributing factors. This highlights a persistent disconnection between technological innovation and effective patent utilization. On the one hand, enterprises should adopt a "use-oriented" patent strategy by strengthening the market alignment of R&D initiatives and avoiding trend-following or redundant patent applications. On the other hand, it is essential to establish a scientific and comprehensive technology evaluation system to enhance the efficiency of patent screening, improve technological maturity, and promote the effective transformation of high-potential innovations. Our methodology already incorporates such an approach through the TOPSIS-based quality assessment, which distinguishes patents with strong commercial potential from those with limited relevance. Building on this, firms should prioritize resource allocation to patents that not only score highly on quality indicators but also exhibit positive legal-event activity (e.g., assignment, licensing, pledge), as these proxies suggest actual or latent commercialization pathways. Furthermore, strengthening collaboration among industry, academia, and research institutions, and encouraging multi-stakeholder joint innovation, can help bridge the gap between original technologies and commercially viable products. Such collaboration should also be oriented around the IPC hotspots identified in our analysis—namely A43B (footwear innovations), A41D (protective apparel), and A63B (knee protectors)—so that joint innovation directly targets the technological areas with the highest strategic value. This recommendation directly reflects our finding that co-assignee and co-citation networks in this domain remain thin and fragmented, indicating limited knowledge exchange and weak linkages between patent creation and industrial application. By fostering collaborative patenting and joint development in emerging technological areas, China can accelerate the transition of running protective equipment from patent documents into market-ready products. Ultimately, this evidence-based approach facilitates the transformation of patent value from passive "paper reserves" into active "industrial capital." Rather than relying on patent counts alone, Chinese enterprises can improve market competitiveness, innovation return on investment, and technological leadership by leveraging the legal-status evidence (validity rates, survival periods), IPC hotspot trajectories, and quality indicators identified in this study.

# Appendix

## A. Search Queries

Patent data were retrieved from the Patsnap Global Patent Database, with the search completed on Jan 15, 2025. The query strategy combined keyword searches across the title, abstract, and claims fields with restrictions based on IPC classifications. The core Boolean expression was as follows:

(TI/AB/CL: ("protective running gear" OR "running protective gear" OR "running shoe" OR "sports footwear" OR "knee pad" OR "knee brace" OR "waist support" OR "waist protector" OR "sports protective clothing" OR "compression wear")) AND IPC: (A43B OR A41D OR A63B) AND PD: [1898-01-01 TO 2023-12-31] AND (LANGUAGE: ("Chinese" OR "English"))

To avoid the inclusion of irrelevant records, the following terms were systematically excluded:

NOT TI/AB/CL: ("industrial safety shoe" OR "work boot" OR "steel toe" OR "ski boot" OR "mountaineer boot" OR "orthopedic" OR "rehabilitation brace" OR "prosthe" OR "medical")

This ensured that the retrieved patents were focused exclusively on technologies related to running protective equipment.

## B. IPC Classifications

To refine the search scope and enhance precision, the following IPC classes and subclasses were used as a whitelist. These codes correspond to the technological domains most relevant to running protective equipment:

A43B–Footwear
A43B5/06: Running shoes; track shoes
A43B5/00: Sports footwear
A43B13/12: Multi-layer soles composed of different materials
A43B13/14: Sole structural shape features
A43B13/18: Elastic soles
A43B13/20: Inflatable soles
A41D–Protective clothing
A41D13/05: Garments designed to protect specific body parts
A41D13/06: Knee pads or foot protectors
A63B–Sporting goods and protective devices
A63B71/12: Protective devices for the body or legs

The selection of these subclasses was validated through iterative testing and manual verification to ensure their relevance to the study domain.

## C. Exclusion Criteria

During the manual validation process, patents falling into the following categories were excluded:

① Industrial or occupational footwear, such as safety boots, steel-toe boots, and mining boots.

② Outdoor equipment not specific to running, such as ski boots and mountaineering boots.

③ Medical or rehabilitative devices, including orthopedic braces, postoperative knee supports, prosthetics, and diabetic foot devices.

④ Non-wearable protective structures, such as running track surfaces or exercise machines.

These exclusion rules were implemented to ensure that the final data set accurately represented patents directly relevant to running protective equipment.

## D. Data Cleaning and Processing

To guarantee data quality, reproducibility, and analytical validity, a multi-step cleaning protocol was implemented:

① Family aggregation (INPADOC): patents were consolidated at the earliest priority level, with family representatives selected to prevent double-counting across jurisdictions or languages.

② Applicant name standardization: corporate entities were harmonized (e.g., "Nike Inc.", "Nike Innovate C.V." consolidated under Nike). Variants, abbreviations, and historical name changes were merged (e.g., "Adidas AG" vs. "Adidas International"). Individual applicants were standardized using the "Last name, First name" format.

③ Duplicate removal: records with identical publication numbers across multiple languages were deduplicated. Records missing legal status or IPC classifications were excluded unless such data could be recovered from other family members.

④ Manual screening: Each patent abstract and claim was independently reviewed by two coders. Discrepancies were resolved by consensus. This manual inspection was essential to eliminate false positives and to ensure the data set was exclusively composed of patents concerning running protective equipment.

## Acknowledgments

The authors are grateful to the Staff members of Physical Education Department of Wuxi Taihu University.

## Author contributions

**Conceptualization:** Qiong Shen.

**Resources:** Lei Qiang.

**Writing – original draft:** Yao Liu.

**Writing – review & editing:** Yao Liu.

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
