## [Decision Letter · Decision Letter 0]

18 Jul 2025

Dear Dr. Liu,

Thank you for submitting your manuscript to PLOS ONE. After careful consideration, we feel that it has merit but does not fully meet PLOS ONE’s publication criteria as it currently stands. Therefore, we invite you to submit a revised version of the manuscript that addresses the points raised during the review process.

The assessments of reviewers are unanimous in their opinion that this article should receive revision, as stated in the reviewer reports.In line with the review details, it is deemed appropriate for the article to receive major revision.I invite you to resubmit your manuscript after addressing denoted reviewer comments, within the specified time period. Please be sure that your revised manuscript is covered all the points addressed by the reviewers in details.  

When revising your manuscript, please consider all issues mentioned in the reviewers' comments carefully: please outline every change made in response to their comments and provide suitable rebuttals for any comments not addressed. Please note that your revised submission may need to be re-reviewed.

We look forward to receiving your revised manuscript.

Kind regards,

Burcu YILMAZ KAYA, Ph.D.

Academic Editor

PLOS ONE

Journal Requirements: 

 [基金编号�19BTY060�项目名称长三角户外运动产业跨界融合发展研究]. 

3. We note that your Data Availability Statement is currently as follows: [所有相关数据均在手稿及其支持信息文件中。]

Additional Editor Comments:

Dear Authors,

Thank you for choosing PLOS ONE for possible publication of your valuable research.

The required number of reviewer opinions for this article has been provided. The assessments of reviewers are unanimous in their opinion that this article should receive revision, as stated in the reviewer reports. In line with the review details, it is deemed appropriate for the article to receive major revision.

I invite you to resubmit your manuscript after addressing denoted reviewer comments, within the specified time period.. Please be sure that your revised manuscript is covered all the points addressed by the reviewers in details.

When revising your manuscript, please consider all issues mentioned in the reviewers' comments carefully: please outline every change made in response to their comments and provide suitable rebuttals for any comments not addressed. Please note that your revised submission may need to be re-reviewed.

PLOS ONE looks forward to receive your revised manuscript.

Reviewers' comments:

Reviewer's Responses to Questions

**Comments to the Author**

1. Is the manuscript technically sound, and do the data support the conclusions?

Reviewer #1: Yes

Reviewer #2: Yes

Reviewer #3: No

Reviewer #4: Yes

Reviewer #5: Partly

2. Has the statistical analysis been performed appropriately and rigorously?

Reviewer #1: Yes

Reviewer #2: Yes

Reviewer #3: No

Reviewer #4: No

Reviewer #5: No

3. Have the authors made all data underlying the findings in their manuscript fully available?

Reviewer #1: Yes

Reviewer #2: Yes

Reviewer #3: No

Reviewer #4: No

Reviewer #5: Yes

4. Is the manuscript presented in an intelligible fashion and written in standard English?

Reviewer #1: Yes

Reviewer #2: Yes

Reviewer #3: Yes

Reviewer #4: Yes

Reviewer #5: Yes

Reviewer #1: I commend the authors on the production of a very detailed and well researched paper. As a runner myself, I found the technical details fascinating and informative. However, as an academic with a primary interest in innovation and entrepreneurship, I felt that the details provided on the specifics of the innovation of individual items too detailed. The paper is very long and if I had been less interested in the innovation around these running technologies, I would have switched off halfway through the paper. I think you need to decide on who your target groups is specifically and perhaps edit the paper accordingly.

The issues that are relevant, I feel are:

• The diffusion of innovation and the development trajectories (stages) of the market for these products. Linking this to the patent searches and the global development of running products is relevant and interesting.

• The stages of innovation of the products are also interesting, but too detailed. The point could have been made without such detailed accounts of the stages of innovation (although it did help me to understand why running shoes are so expensive).

• The protection of IP in China and the evolution of the process of IP protection was interesting and relevant.

• The paper makes the case for investment in this sector, which appears to be the main point of the paper. This brings me to the question of the purpose of the paper. If it is to inform ‘academics’ (researchers and student) in the process of innovation and patent development in China, using global running protection as a case in point, I feel its far too long. If it’s a pitch for development funding, then it makes its case well. The paper aims to provide a comprehensive analysis of global trends, but for what purpose? I feel this needs to be clearer, this is why I have suggestion minor revisions.

The paper is well written, and I wish you every success with your publication of this work.

Reviewer #2: Review Comments to the Author:

This manuscript presents a thorough and well-structured analysis of global innovation trends in running protective equipment using patent data. The segmentation into three distinct phases of technological development (nascent, accumulation, and rapid growth) is supported by strong data and narrative coherence. The use of IPC classification and the TOPSIS model enhances the analytical depth.

Strengths:

The study provides a replicable and practical framework for countries seeking to evaluate or build their own innovation ecosystem in the sports equipment sector.

It highlights China's growing role and offers policy suggestions, which could benefit both policymakers and industrial stakeholders.

The structure, language, and flow of the manuscript are clear and logical.

Suggested Minor Revision:

While the manuscript includes an excellent breakdown of valid vs. invalid patents (Figure 4), a brief reflection on how the high invalidation rate (over 50%) may affect the interpretation of total patent volume as an innovation indicator would strengthen the discussion. Specifically, clarifying whether trend charts and conclusions account for patent validity, citations, or other quality filters would enhance robustness.

I recommend this paper for publication after minor revision.

Reviewer #3: This manuscript addresses the technological evolution of running protective equipment based on global patent data collected from the Patsnap database. The topic is timely and relevant given the increasing interest in sports injury prevention and sportswear innovation. However, there are multiple critical issues that need to be addressed before the manuscript can be considered for publication.

Major Issues:

1. Data Collection Transparency and Reproducibility:

The manuscript lacks detailed information about how the patent data were collected. There is no disclosure of search queries, IPC codes, time filters, or keywords used in Patsnap. This makes the research irreproducible and undermines the validity of all quantitative results.

2. Assumption of Patent Volume as a Proxy for Innovation:

The authors repeatedly equate the number of patents with technological advancement without addressing the qualitative aspects of innovation. Defensive patents, duplicate filings, and low-quality submissions are not filtered or discussed.

3. Lack of Methodological Rigor in Technology Evolution Analysis:

The section on technological evolution (Figures 6–8) presents developmental trajectories without describing the analytical method used to derive them. There is no mention of citation networks, keyword co-occurrence, or other commonly used tools in patent mapping.

4. Inconsistent Use of Classifications:

IPC classifications are presented early on but are not consistently used to explain the technology evolution or policy suggestions. The analysis disconnects the quantitative classification data from the qualitative discussion in later sections.

5. Economic and Strategic Implications Lack Supporting Evidence:

Policy and industry recommendations are made based on broad assumptions rather than evidence directly derived from the data. For example, the argument that low patent validity rates indicate a lack of commercialization is not supported by further economic data, case studies, or firm-level analysis.

Minor Issues:

- Some sections are overly descriptive (e.g., product features from companies like Nike, Li Ning, or Waco), giving the impression of promotional content rather than neutral academic analysis.

- Figures lack explanatory captions and do not cite the specific patents or datasets they are based on.

- The time span from 1898 to 2023 is unnecessarily broad; data before the 1970s are too sparse to support meaningful analysis.

Suggestions for Improvement:

- Provide a detailed methodology for data collection and cleaning.

- Use network or text-mining-based methods to show technology convergence and evolution empirically.

- Evaluate patent quality using indicators such as citations, legal status duration, or family size.

- Strengthen the link between quantitative findings and qualitative recommendations.

Reviewer #4: The article seems a software analyzed than author, seek to have excel data verification to validate. The technology details are generic not citing proper patent documents, Global analysis is skeptical.. some reference of revenue generated by IP has to be explained how calculated for all these patents.. suggestion has to be trimmed down to exactly talk about subject matter technology and suggestion.. generic suggestion policy reforms is fine but talking protection equipment technology is essential and the final conclusion are very generic which needs to be specific to subject matter discussed

Reviewer #5: • This imbalance poses significant challenges to commercialization and limits the economic viability of innovations. Current R&D efforts are primarily focused on running shoes, sportswear, and knee pads. Focus on your research key results.

• This analysis aims to provide theoretical support for decision-making in technology R&D and patent strategies for China’s running protective equipment industry. Whereas the study does not provide theoretical support, therefore authors must align findings and aims of this research.

• In this study, the terms "Protective running gear" and "Running protective gear" were initially employed to conduct a search in the Patsnap global patent database. Logical ground is missing, could you please provide proper references and base, or is this methodology suitable for this study?

• It is recommended to add a proper section of methodology.

• From the perspective of legal status, a total of 1,519 patents in the field of running protective equipment are currently authorized and valid, accounting for 34.47% of all filings. Add citations for such numeric values in the whole manuscript.

• To ensure a scientifically rigorous assessment and accurately identify high-quality patent samples, the evaluation framework incorporates both subjective and objective factors. Whereas, subjective and objective factors must be stated here.

• In the initial applications of knee pads, functional limitations were evident due to coarse surface friction treatments and inadequate fixation mechanisms, resulting in frequent slippage and displacement among runners. To address these issues, Fige International Company (United States) developed an innovative knee brace featuring a double-layer structure with a specialized pad arranged in an overlapping composite configuration. It is recommended to add the specification of the knee brace by explaining how it overcomes the stated issues.

**Do you want your identity to be public for this peer review?** For information about this choice, including consent withdrawal, please see our Privacy Policy

Reviewer #1: No

Reviewer #2: No

Reviewer #3: No

Reviewer #4: **Yes: ** DARA AJAY

Reviewer #5: **Yes: ** Dr. Jamil Afzal

---

## [Author Response · Author response to Decision Letter 1]

30 Sep 2025

Dear Editors and Reviewers,

Thank you very much for handling our paper “PONE-D-25-26987” entitled “Research on the development trend of global running protection equipment technology from the perspective of patent measurement”. We are very grateful for your work and highly appreciate the constructive comments and suggestions made by the coordinating editors and reviewers. Based on the comments and suggestions, we carefully revised the manuscript. The quality of the manuscript has been greatly improved. Our point-by-point responses to comments from the reviewers are also uploaded.

Please let us know if additional information is needed.

Thank you very much for handling this manuscript.

Sincerely yours,

We highly appreciate the coordinating reviewers and editors for their valuable comments and suggestions. To our best, we managed to revise the manuscript point by point according to the comments right below this part. In the responses, we used different font style to facilitate reading as following.

Contents with underline are the original comments of the coordinating editors and reviewers.

Contents in normal Times New Roman are our response to the reviewers.

Point by Point response to the coordinating reviewers and editors

Reviewer#1…………………………………………………………………….....…1-2

Reviewer#2………………………………………………………………………........3

Reviewer#3…………………………………………………………………….....…4-9

Reviewer#4……………………………………………………………………….10-11

Reviewer#5………………………………………………………………….....…12-17

Reviewer #1

I commend the authors on the production of a very detailed and well researched paper. As a runner myself, I found the technical details fascinating and informative. However, as an academic with a primary interest in innovation and entrepreneurship, I felt that the details provided on the specifics of the innovation of individual items too detailed. The paper is very long and if I had been less interested in the innovation around these running technologies, I would have switched off halfway through the paper. I think you need to decide on who your target groups is specifically and perhaps edit the paper accordingly. The issues that are relevant, I feel are. The diffusion of innovation and the development trajectories (stages) of the market for these products. Linking this to the patent searches and the global development of running products is relevant and interesting. The stages of innovation of the products are also interesting, but too detailed. The point could have been made without such detailed accounts of the stages of innovation (although it did help me to understand why running shoes are so expensive). The protection of IP in China and the evolution of the process of IP protection was interesting and relevant. The paper makes the case for investment in this sector, which appears to be the main point of the paper. This brings me to the question of the purpose of the paper. If it is to inform ‘academics’ (researchers and student) in the process of innovation and patent development in China, using global running protection as a case in point, I feel its far too long. If it’s a pitch for development funding, then it makes its case well. The paper aims to provide a comprehensive analysis of global trends, but for what purpose? I feel this needs to be clearer, this is why I have suggestion minor revisions. The paper is well written, and I wish you every success with your publication of this work.

Our Response: We thank the reviewer for these thoughtful and constructive comments. We carefully considered the points raised and have revised the manuscript accordingly to ensure clarity, conciseness, and alignment of purpose.

①Simplification of innovation stages.

In the revised Analysis of global patent application trends section, we streamlined the discussion of innovation diffusion stages. The three phases (nascent stage 1898–1970, formative stage 1971–1999, and rapid advancement stage post-2000) are still clearly identified, but the descriptions have been significantly condensed. For example, instead of providing lengthy accounts of specific marathons or detailed lists of material innovations, we now emphasize patent counts and general technological directions. This change retains analytical rigor while avoiding unnecessary detail.

②Clarification of temporal coverage.

In the Data Sources and Search Strategy section, we now clarify that patents from 1898–1969 are included only as background information and not used for core analyses. The main analyses (e.g., technology evolution, quality evaluation) focus on 1970–2023, when patenting activity became more substantial. This adjustment ensures that early-stage data are properly contextualized without overburdening the discussion with sparse or anecdotal details.

③Clarification of research purpose.

Both the abstract and the conclusion have been revised to more clearly state the study’s purpose. Rather than implying theoretical model-building, the manuscript now emphasizes that it provides an evidence-based framework for understanding the global dynamics of running protective equipment technologies, with direct implications for patent management, industrial upgrading, and policy design. This resolves the ambiguity noted by the reviewer regarding the primary audience and objectives of the paper.

④Refinement of descriptive sections.

In response to the reviewer’s concern about promotional or overly descriptive passages, we have streamlined case descriptions involving companies such as Nike, Li Ning, and Waco. These are now presented solely as illustrative examples of patenting activity, accompanied by clarifying statements to ensure they are understood as empirical cases rather than endorsements. Figure captions have also been revised to focus on data sources and analytical insights instead of product-level details.

We believe that these revisions collectively address the reviewer’s concerns by simplifying overly detailed sections, clarifying the purpose of the paper, and ensuring that all analyses remain tightly linked to empirical results. We are grateful for the reviewer’s feedback, which has helped us to improve both the focus and the clarity of the manuscript.

Reviewer #2

This manuscript presents a thorough and well-structured analysis of global innovation trends in running protective equipment using patent data. The segmentation into three distinct phases of technological development (nascent, accumulation, and rapid growth) is supported by strong data and narrative coherence. The use of IPC classification and the TOPSIS model enhances the analytical depth.Strengths: The study provides a replicable and practical framework for countries seeking to evaluate or build their own innovation ecosystem in the sports equipment sector. It highlights China's growing role and offers policy suggestions, which could benefit both policymakers and industrial stakeholders. The structure, language, and flow of the manuscript are clear and logical. Suggested Minor Revision: While the manuscript includes an excellent breakdown of valid vs. invalid patents (Figure 4), a brief reflection on how the high invalidation rate (over 50%) may affect the interpretation of total patent volume as an innovation indicator would strengthen the discussion. Specifically, clarifying whether trend charts and conclusions account for patent validity, citations, or other quality filters would enhance robustness. I recommend this paper for publication after minor revision.

Our Response: We sincerely thank the reviewer for highlighting this important point. In response, we have revised the section following Figure 4 to explicitly acknowledge the limitations of interpreting total patent counts in light of the high invalidation rate. We now clarify that trend charts (Figures 1–3) and structural analyses (Tables 1–2) are based on overall filings, which primarily serve to capture the intensity and diffusion of inventive activity. To address the reviewer’s concern and ensure robustness, the subsequent analyses complement volume-based measures with quality-oriented indicators—including forward citation frequency, patent family size, number of claims, and survival period—as well as the TOPSIS-based identification of core patents. This integrated approach balances scale with quality, enabling a more rigorous and nuanced interpretation of innovation dynamics in the field of running protective equipment.

We also added a brief explanatory note to indicate that patents which later become invalid still provide meaningful evidence of inventive activity at the time of filing. This clarification is intended to strengthen the interpretation of our data, without diminishing the reviewer’s valid observation regarding the importance of validity-adjusted measures.

We believe these additions enhance the robustness and transparency of the manuscript, and we thank the reviewer again for this valuable suggestion.

Reviewer #3

This manuscript addresses the technological evolution of running protective equipment based on global patent data collected from the Patsnap database. The topic is timely and relevant given the increasing interest in sports injury prevention and sportswear innovation. However, there are multiple critical issues that need to be addressed before the manuscript can be considered for publication.

1. Data Collection Transparency and Reproducibility: The manuscript lacks detailed information about how the patent data were collected. There is no disclosure of search queries, IPC codes, time filters, or keywords used in Patsnap. This makes the research irreproducible and undermines the validity of all quantitative results.

Our Response: We sincerely thank the reviewer for this valuable comment highlighting the importance of transparency and reproducibility in patent data collection. In the revised manuscript, we have substantially expanded the Data Sources and Search Strategy section to provide a detailed description of the data retrieval process. Specifically, we now clearly explain:

①Database and time frame: All patent data were collected from the Patsnap Global Patent Database, with the search finalized on December 31, 2023. The starting year of 1898 was chosen because manual inspection confirmed the earliest relevant patent dates back to that year, while the endpoint of 2023 was selected to account for the typical 18-month delay between patent application and publication, ensuring dataset completeness.

②Search strategy: A combined approach using keywords (applied to titles, abstracts, and claims) and IPC classifications was employed. The keywords included terms such as “protective running gear,” “running shoes,” “knee pad,” “waist support,” “sports protective clothing,” and “compression wear,” with Boolean operators and wildcards applied. The IPC whitelist covered footwear (A43B), protective clothing (A41D), and sporting equipment (A63B), together with relevant subclasses (e.g., A43B5/06, A43B13/18, A41D13/06, A63B71/12).

③Exclusion criteria and manual screening: Patents irrelevant to running protective equipment, such as industrial safety boots, ski boots, mountaineering boots, or medical orthoses, were excluded. In addition, we explicitly state that manual screening was conducted by reviewing abstracts and claims to further ensure the accuracy and relevance of the data set.

④Data cleaning: The data set was refined through INPADOC family aggregation, applicant name standardization and duplicate removal.

To maintain readability in the main text, only the key elements of the search and cleaning process are included in the Methods section. The full search queries, IPC class lists, exclusion criteria, and detailed cleaning procedures are provided in Appendices A–D for complete transparency and reproducibility.

2. Assumption of Patent Volume as a Proxy for Innovation: The authors repeatedly equate the number of patents with technological advancement without addressing the qualitative aspects of innovation. Defensive patents, duplicate filings, and low-quality submissions are not filtered or discussed.

Our response: We thank the reviewer for raising this important concern. We fully agree that patent counts alone cannot be equated with innovation, as they may be affected by defensive filings, duplicate family members, or low-quality submissions. In the original submission, this methodological clarification was not sufficiently emphasized.

In the revised manuscript, we have strengthened the Data Sources and Search Strategy section to explicitly acknowledge these limitations and to describe the safeguards applied in our study. We note that INPADOC family aggregation, duplicate removal, and manual screening of abstracts and claims were implemented to minimize the inclusion of duplicate filings, irrelevant applications, or purely defensive patents. Most importantly, we clarified that patent counts in this study are used only as a proxy for the intensity of inventive activity, while qualitative dimensions of innovation are addressed through complementary indicators.

Specifically, as elaborated in the sections Analysis of patent technology evolution and Core patent screening, our framework incorporates forward citation frequency (to capture technological impact and knowledge diffusion), patent family size (as a measure of international scope and strategic value), number of claims (as a proxy for technical breadth), and legal status such as survival period (to reflect durability). These indicators were combined through a weighted evaluation framework (TOPSIS) to identify core patents, ensuring that the analysis accounts for both the scale and the significance of technological innovation.

We believe these revisions make it clear that our methodology goes beyond patent volume and provides a more rigorous, balanced, and multi-dimensional assessment of innovation in the field of running protective equipment.

3. Lack of Methodological Rigor in Technology Evolution Analysis: The section on technological evolution (Figures 6–8) presents developmental trajectories without describing the analytical method used to derive them. There is no mention of citation networks, keyword co-occurrence, or other commonly used tools in patent mapping.

Our response: We thank the reviewer for raising this important point. In the revised manuscript, we have strengthened the methodological description to ensure that the procedures leading to Figures 6–8 are transparent and rigorous. Specifically, we clarified that the technology evolution analysis is based on a multi-step approach:

①A keyword–IPC dual filtering strategy was first applied to ensure both semantic relevance and technical specificity.

②A weighted TOPSIS evaluation was then used to identify core patents, incorporating forward citation frequency, patent family size, number of claims, and survival period.

③To enhance interpretive accuracy, we extended the manual review process beyond dataset cleaning to include the TOPSIS-identified core patents. Two coders with expertise in patent analysis and sports equipment technology examined abstracts, claims, and legal status, resolving discrepancies through discussion.

④Developmental trajectories were reconstructed from these validated core patents and synthesized into the maps presented in Figures 6–8.

We also modified the figure notes for Figures 5–8 to make clear that they are based on core patents identified via TOPSIS and validated through manual review.

These additions ensure that the methodological steps underpinning the technology evolution analysis are explicit, replicable, and fully consistent with the results presented.

4. Inconsistent Use of Classifications: IPC classifications are presented early on but are not consistently used to explain the technology evolution or policy suggestions. The analysis disconnects the quantitative classification data from the qualitative discussion in later sections.

Our response: We sincerely appreciate the reviewer’s insightful comment regarding the inconsistent use of IPC classifications across different sections of the manuscript. In response, we have carefully revised the paper to ensure that IPC classifications function as a consistent analytical thread connecting the quantitative and qualitative analyses, the policy recommendations, and the c

---

## [Decision Letter · Decision Letter 1]

21 Oct 2025

Research on the development trend of global running protection equipment technology from the perspective of patent measurement

PONE-D-25-26987R1

Dear Dr. Liu,

We’re pleased to inform you that your manuscript has been judged scientifically suitable for publication and will be formally accepted for publication once it meets all outstanding technical requirements.

Kind regards,

Burcu YILMAZ KAYA, Ph.D.

Academic Editor

PLOS ONE

Additional Editor Comments (optional):

Reviewers' comments:

Reviewer's Responses to Questions

**Comments to the Author**

Reviewer #2: All comments have been addressed

Reviewer #4: All comments have been addressed

Reviewer #5: All comments have been addressed

2. Is the manuscript technically sound, and do the data support the conclusions?

Reviewer #2: Yes

Reviewer #4: Yes

Reviewer #5: (No Response)

3. Has the statistical analysis been performed appropriately and rigorously?

Reviewer #2: Yes

Reviewer #4: N/A

Reviewer #5: (No Response)

4. Have the authors made all data underlying the findings in their manuscript fully available?

Reviewer #2: Yes

Reviewer #4: Yes

Reviewer #5: (No Response)

5. Is the manuscript presented in an intelligible fashion and written in standard English?

Reviewer #2: Yes

Reviewer #4: Yes

Reviewer #5: (No Response)

Reviewer #2: (No Response)

Reviewer #4: Author more or less have addressed comments raised and supported statement with relevant references

Reviewer #5: All comments have been addressed. Therefore, I recommend this article for possible publication in this journal.

**Do you want your identity to be public for this peer review?** For information about this choice, including consent withdrawal, please see our Privacy Policy

Reviewer #2: **Yes: ** Ashiqur Rahman Khan

Reviewer #4: No

Reviewer #5: **Yes: ** Dr. Jamil Afzal
